# Low-Cost Fluorescence Sensor for Ammonia Measurement in Livestock Houses

**DOI:** 10.3390/s21051701

**Published:** 2021-03-02

**Authors:** Jesper Nørlem Kamp, Lise Lotte Sørensen, Michael Jørgen Hansen, Tavs Nyord, Anders Feilberg

**Affiliations:** 1Environmental Engineering, Department of Biological and Chemical Engineering, Aarhus University, 8200 Aarhus N, Denmark; michaelj.hansen@bce.au.dk (M.J.H.); tavs.nyord@bce.au.dk (T.N.); af@bce.au.dk (A.F.); 2Department of Environmental Science, Aarhus University, 4000 Roskilde, Denmark; lls@envs.au.dk

**Keywords:** ammonia, NH_3_, fluorescence sensor, pigs

## Abstract

Measurements of ammonia with inexpensive and reliable sensors are necessary to obtain information about e.g., ammonia emissions. The concentration information is needed for mitigation technologies and documentation of existing technologies in agriculture. A flow-based fluorescence sensor to measure ammonia gas was developed. The automated sensor is robust, flexible and made from inexpensive components. Ammonia is transferred to water in a miniaturized scrubber with high transfer efficiency (>99%) and reacts with o-phthalaldehyde and sulfite (pH 11) to form a fluorescent adduct, which is detected with a photodiode. Laboratory calibrations with standard gas show good linearity over a dynamic range from 0.03 to 14 ppm, and the detection limit of the analyzer based on three-times the standard deviation of blank noise was approximately 10 ppb. The sampling frequency is 0.1 to 10 s, which can easily be changed through serial commands along with UV LED current and filter length. Parallel measurements with a cavity ring-down spectroscopy analyzer in a pig house show good agreement (R^2^ = 0.99). The fluorescence sensor has the potential to provide ammonia gas measurements in an agricultural environment with high time resolution and linearity over a broad range of concentrations.

## 1. Introduction

Ammonia (NH_3_) emissions to the atmosphere originate mainly from agricultural activities, and only minor fractions originate from combustion, transport, industry and waste [1]. The interest in NH_3_ arises from the effects on human health and the environment as NH_3_ travels downwind from the source to have an impact on areas separated from the source [2]. The local environment can be affected by acidification of the terrestrial environment, eutrophication of local waters and affecting the biodiversity in terrestrial ecosystems [3]. The formation of particles is enhanced by NH_3_ in the formation of ammonium aerosols [4,5] that have longer atmospheric lifetimes compared to NH_3_, which allow for long-range transport and ultimately deposition further away from the source. NH_3_ has a significant role in small particle formation [6], which has a negative influence on human health [7]. Furthermore, as a precursor to the potent greenhouse gas nitrous oxide, NH_3_ has indirect effect on climate change [3].

It is highly important to have reliable measurement techniques for NH_3_ in an agricultural environment because the agricultural sector emits most of the anthropogenic NH_3_ [8]. Concentration measurements of NH_3_ is a cornerstone in emission determination and more available measurement methods will increase the knowledge on NH_3_ emissions. In addition, reliable and cost-effective methods are needed to document, develop, and optimize NH_3_ mitigation technologies in agriculture, e.g., validation of air cleaners for livestock buildings or injection methods for field application of slurry.

The ternary reaction between o-phthaldialdehyde (OPA), a reducing agent, and NH_3_ or a primary amino acid produces an intense fluorescent product [9], which can be used to detect NH_3_ or amino acids [10,11]. In the reaction, NH_3_, sulfite, and OPA form 1-sulfonate isoindole [12]. Mercaptoethanol, boronhydride [9] and sulfite [13] has been used as reducing agent. An automated flow-injection method used sulfite as a reducing agent with OPA to measure NH_3_/NH_4_^+^ with fluorescence or absorption detection [11]. The use of OPA to determine NH_3_/NH_4_^+^ with fluorescence has been investigated in many studies and different minor adjustments have perfected it to suit specific tasks, e.g., detection of ammonium ion in marine and freshwater ecosystems [14]. The method proposed by Genfa and Dasgupta [11] is sensitive and selective for NH_3_ [11,14]. The high solubility of NH_3_ in water makes the fluorometric measurement technique promising for online measuring as done by Ohira et al. [15] who developed an automated system to measure NH_3_ gas in indoor air with a detection limit of 0.9 ppb. After transferring NH_3_ from air to water with a scrubber, the aqueous NH_3_ reacts with sulfite and OPA and the product of this reaction (1-sulfonate isoindole) is fluorescent, which is measured with a photodetector. The product (1-sulfonate isoindole) has an excitation maximum of 362 nm, whereas the fluorescent emission has a maximum of 422 nm, thus a long-pass filter can reduce scattered light from the excitation source at the detector [15]. The sensor by Ohira et al. [15] is robust in an indoor environment with relatively low concentrations and measurement time of 90 s per sample, thus lower measurement time and test in high concentration environments like in agriculture is needed.

Several measurement techniques for measuring NH_3_ are available [16], and measurements in an agricultural environments are routinely performed with e.g., cavity ring-down spectroscopy (CRDS) [17], AMANDA [18], PAS [19], miniDOAS [20], PTR-MS [21], ALPHA samplers [22], Leuning tubes [23] and impingers [24]. Most of these techniques are expensive (e.g., PTR-MS and CRDS) and/or labor intensive (e.g., impingers, ALPHA samplers, and Leuning tubes) while others are prone to systematic errors [19].

The aim of this study was to develop a low cost sensor for continuous measurements of NH_3_ in the air of an agricultural environment with relatively high concentrations (>10 ppm) based on absorption of NH_3_ in water and subsequent fluorescence detection. The fluorescence sensor is a proof-of-concept for a small, robust, flexible, and inexpensive sensor compared to other available measurement techniques. The sensor is subsequently tested for measuring NH_3_ in livestock houses.

## 2. Materials and Methods

### 2.1. Fluorescence Sensor System

The fluorescence sensor system consists of four parts (1) scrubber sampling, (2) pumps with mixing of reagents and sample, (3) heating and (4) detector. The liquid pumps are liquid dispense pumps of the solenoid type (SMC Denmark, Horsens, Denmark) and the air pump is a diaphragm pump (Boxer GmbH, Ottobeuren, Germany). A schematic of the system is shown in Figure 1. Sample air is pulled through the scrubber with a flow of 2.4 L min^−1^ and the water flow to the scrubber is 4.9 mL min^−1^. The water flow pulling away from the scrubber is just slightly higher to avoid the build-up of water inside the scrubber. Inline after the scrubber a T-piece (debubbler) is inserted. The debubbler is oriented horizontally allowing air to escape upwards with the overflow going to waste. A small amount of water is sucked from the underside of the de-bubbler ensuring a minimum of air bubbles in the sample. The flow from the debubbler is 0.22 mL min^−1^. Sulfite and OPA are pumped with 0.22 mL min^−1^, and mixed with the sample water containing NH_3_ before heating to 75 °C. The temperature controller has a precision of 0.1 °C, and the heating coil has an inner diameter (i.d.) of 0.75 mm and a length of 0.5 m to ensure sufficient heating of the chemicals. Finally, the heated aqueous solution reaches the detector. The reagent and water containers are sealed and connected to a charcoal filter in order to equalize the pressure without contaminating the headspace air above the liquids.

### 2.2. Scrubber

The scrubber consists of a glass cylinder with outlet and inlet for both water and air in countercurrent flow with water entering in the top and air in the bottom. A glass blasted cylinder was inserted inside the glass cylinder ensuring a large surface area for the laminar flow of water. The outer diameters of the two glass cylinders are 26 mm and 22 mm with 1 mm wall thickness. The length of the glass blasted inner cylinder is 12.5 cm and the distance between the two air inlets is 13.5 cm. The total length is 20 cm. See Figure A1 for a picture of the scrubber. 

### 2.3. Chemicals and Instruments

The following chemicals were used for reagents: o-phthalaldehyde (OPA, ≥97%, Sigma-Aldrich, St. Louis, MO, USA), sodium hydroxide (VWR, Leuven, Belgium), ethanol (VWR, Fontena-sous-Bois, France), sodium sulfite (Merck, Darmstadt, Germany), disodium phosphate (≥99%, Sigma-Aldrich, St. Louis, MO, USA), formaldehyde (Sigma-Aldrich, St. Louis, MO, USA), and Millipore water.

Ultrapure Millipore water is the absorbing agent in the scrubber, which is mixed with 10 mM OPA and 3 mM sodium sulfite in 0.19 M phosphate buffer (pH 11). OPA is prepared by dissolving 1.34 g/L OPA in 1/4 ethanol and 3/4 ultrapure water. Phosphate buffer is prepared by adding 26.8 g/L disodium phosphate to ultrapure water and adjusting the pH to 11 with sodium hydroxide. The sodium sulfite (0.378 g/L) were added to the phosphate buffer. The reagents were prepared in several batched during the laboratory calibrations and the pig house measurements.

Measurements of NH_3_ with a CRDS analyzer from Picarro model G2103 (Picarro Inc., Santa Clara, CA, USA) was compared to NH_3_ measured with the fluorescence sensor presented here. This specific CRDS-instrument has been proven reliable in an agricultural environment [17].

The following gases were used: NH_3_ in N_2_ (Air Liquide, Taastrup, Denmark) with different concentrations ranging from 5 ppm to 5% with uncertainty from 3% to 5% and N_2_ (Air Liquide, Taastrup, Denmark). Mass flow controllers (Bronkhorst, AK Ruurlo, The Netherlands) were used for a dynamic dilution system to obtain a range of NH_3_ concentrations during the laboratory calibrations.

### 2.4. Fluorescence Detector

The fluorescence detector consists of a photodiode and an UV LED, and these two are mounted in a custom-made flow cell. The photodiode and LED are installed in mounts to secure perfect alignment and centering of the components to obtain as precise and reproducible measurements as possible. Figure 2 shows technical drawings with placement of mounts, ball lenses, and flow channel. Figure A2 shows a picture of the detector. Each of these two mounts sits on top of a 3.0 mm sapphire ball lens. The mounts are tightened with a spring system to avoid deformation, but the mounts still push down the ball lenses.

The excitation light is introduced via a ball lens and the emitted light is transferred to the photodiode via a second ball lens. The distance from the edge of the lens to the glass of the photodiode is only 0.215 mm to avoid loss of light, as the focal point of the lens is 0.224 mm from the edge of the lens. The flow cell is built in polyether ether ketone (PEEK) because PEEK has a good chemical resistance and high temperature tolerance; furthermore, it is relatively tough compared to other plastics, which makes it possible to produce very precise components such as this. An OPT-301 (Burr-Brown, Tuscon, AZ, USA) integrated photodiode and amplifier with a 100 MΩ resistance as external feedback resistor and a decoupled power supply is used in the setup to give a highly sensitive response of the analyzer. An ultraviolet light-emitting diode (UV LED, Roithner Lasertechnik, Vienna, Austria, λ_max_ 365 nm, 2 mW at 20 mA) is the excitation light source installed with a 100-Ω resistor. A thin sheet of polyester film is placed directly in front of the photodiode as a long-pass filter with a cut-on wavelength of 400 nm. The transmission is <10% below 390 nm, thus most of the light from the UV LED is removed by the filter.

An averaging filter takes advantage of the processing power of the Teensy, thus all data saved is average values of 1–50,000 measurements, see Appendix A for further description of the setup. Additionally, spikes are removed in the post-processing of the data to make the system more robust to disturbances from e.g., bubbles escaping the debubbler. Spikes are defined as data more than three local scaled median absolute deviations (MAD) from the local median over a defined window length. The analog-to-digital-converter converts the mV signal from the photodiode to a 16-bit signal with the range 0–65,535. All output signals from the sensor will be in the arbitrary unit related to the 16-bit signal.

In the present study, the LED current was set to 800, the sampling time to 5000 ms, the filter to 50,000 points, and the window length for spike detection was 18 points (1.5 min).

### 2.5. Pig House

Over a period of five days, the NH_3_ concentration was measured in the outlet air from four pig houses at Aarhus University, Foulum. The four pig houses each contained two pens with 15 growing-finishing pigs. Each pen was 11 m^2^ (2.4 × 4.6 m^2^) with 2.6 m walls and a flat ceiling. The ventilation system in the pig houses was a negative pressure system with air inlet through a diffuse ceiling and one outlet in the ceiling.

Five PTFE pumps (Capex L2, Charles Austen Pumps Ltd., Byfleet, UK) continuously pulled approximately 4 L min^−1^ of air from each measurement position through heated and insulated PTFE tubes (i.d.: 6 mm) into an insulated room next to the pig houses, where all instruments were placed. The setup was similar to the study by Hansen et al. [25]. The CRDS and fluorescence sensor pulled air from the pressure side of pump and excess air from the pumps were pumped into the measurement room. The CRDS used an automated valve switch between the five measurement positions every six minutes in a continuous cycle resulting in two measurements each hour. The position of the fluorescence sensor was changed manually approximately two times a day between two pig houses and an outside background position. Furthermore, synthetic air was applied five times during the measurements for instrumental background.

## 3. Results

### 3.1. Scrubber

The performance and efficiency of the scrubber was tested with a diluted NH_3_ standard gas. As an example, the scrubber was exposed to a constant NH_3_ gas flow of 9280.4 ± 464 ppb. The outlet air from the scrubber with and without water flow was measured with a CRDS analyzer to investigate the scrubber sampling efficiency. The response to a step change with water flowing in the scrubber was investigated, and it took 16 and 40 s before the concentration measured in the outlet air from the scrubber was below 10% and 5% of the initial concentration, respectively. The concentration was the same in the inlet and outlet without water flowing over the scrubber. It should be noted that it took some time for the system to handle such a change as NH_3_ can be adsorbed to surfaces and it took some time before the water flow was running without disturbances at steady state. After 10 min, the concentration was below 40 ppb, corresponding to less than 0.5% of the concentration initially exposed to the scrubber. Tests with NH_3_ inlet concentrations of 1–10 ppm and water pH from 3.4 to 9.9 yield an average collection efficiency of 99.82 ± 0.15% (N = 12). Overall, the scrubber removes >99% of NH_3_ from the air stream.

### 3.2. Laboratory Calibrations

The laboratory calibrations were performed on six different days with a dynamic dilution system to have a wide range of concentrations in the calibration. The non-zero calibration concentrations ranged from 30 ppb to 12 ppm and the calibrations showed high degree of linearity with small differences between the days of calibration and gas standards. Furthermore, a difference in response was observed for different batches of reagents. The regression coefficients were above 0.99 for all calibrations. Figure 3 shows the calibration curve for the smallest concentrations used in the calibration (30–200 ppb). The limit of detection (LOD) is determined as three times the standard deviation of the blank divided by the slope of the regression assuming the standard deviation of the blank is similar to the standard deviation of samples near the detection limit. The standard deviation from the blank measurements and the calibration curve shown in Figure 3 was used to determine the LOD, which yields LOD = 10.2 ppb. Using the standard deviation of blank measurements over six different days, the LOD is in the range of 7.1–17.5 ppb using the calibration curve in Figure 3.

The effect of changing the UV LED current was investigated at a constant concentration of 14 ppm because the maximum expected concentration in the pig house is within 10 and 15 ppm [23]. As seen in Figure 4, there is a high degree of linearity between the set point of the UV LED current and the sensor output. The output was saturated above UV LED current 800, thus this value was used in the calibrations. However, examination of subsequent calibrations demonstrate that some measurements of 14 ppm standard gas were saturated. Hence, with the presented configuration of the sensor, the dynamic range is from 0.03 to approximately 14 ppm.

### 3.3. Pig House Measurements

The NH_3_ concentration was measured with a CRDS analyzer in five different positions, for six minutes at each position. The fluorescence sensor measures from three of these positions by manually switching inlet position between an outside background and two different pig houses. It takes some time before the concentration is stable after each change of position due to the adsorption of NH_3_ to the surfaces of tubing and the instrument itself, thus only the last three minutes of the CRDS measurements are used at each position. The data from both the fluorescence detector and the CRDS analyzer were averaged over 1 min and only compared when they measured at the same position.

Measurements using synthetic air (NH_3_-free) were performed three times to obtain instrumental background. The coefficient of variation (CV) of the three measurements is 2.3% and the standard deviation of each blank measurement was similar to the blank measurements in the laboratory. Using the propagation of uncertainty with a calibration curve as Figure 3 gives an estimate of the uncertainty in the measured concentration with the sensor [24]. Using sensor output of 12,000, 25,000, 45,000 and 65,535 (i.e., approximately 0.1 ppm, 2 ppm, 6 ppm and a saturated detector) yields relative uncertainties of the measured concentration of less than 0.25%. Similar uncertainties are derived from the comparison in Figure 5 with 0.63% uncertainty for approximately 0.1 ppm and uncertainty of approximately 2 ppm, 6 ppm and saturated detector are all below 0.30%.

The scatter plot of measurements using the CRDS and the fluorescence sensor is shown in Figure 5 with a high degree of correlation for the comparison (>0.99). The laboratory calibration (see Figure 3) is used for the fluorescence sensor. Figure 6 compares the concentration measured by the two instruments over time. There are approximately 12 times more minute mean data points from the fluorescence sensor due to the changing positions of the CRDS. The mean concentrations and CV for each sensor at each measurement position is shown in Table 1.

## 4. Discussion

### 4.1. Laboratory Calibrations

The laboratory calibrations showed a high degree of linearity over a broad concentration range (from the detection limit to 14 ppm). There are some minor differences between different days, which indicates differences in the sensitivity of the system. Different batches of the reagents were used for the calibrations, which can have a significant effect on the fluorescence intensity because the desired fluorogenic reaction have clear fluorescence maximum for reaction temperature, pH of buffer, OPA concentration, and sulfite concentration [11]. Therefore, even small differences in OPA or sulfite concentration or pH can change the fluorescence signal of the detector and thereby the sensitivity, which makes it important to calibrate the system with the reagents used in later measurements.

A detector system based on the same fluorescence method as the one presented in the present study obtained an LOD of 0.9 ppb [15], which is somewhat lower than the LOD measured in the present study (10.2 ppb). However, there were some major differences between the systems. The study by Ohira et al. [15] showed a CV of 3.2% for measurement over 24 h of an NH_3_ standard (53 ppb), the sampling frequency was 90 s per sample, and all calibrations are performed on gas concentrations below 75 ppb [15]. All calibrations conducted in the present study lasted for less than 1.5 h and only the last 5 min of steady measurements were used, which gave a mean CV for blanks (number of sample set, N = 5) of 0.18%, calibration concentration above 0 and below 1 ppm gave mean standard deviation of 0.55% (N = 5), and calibration concentrations above 1 ppm gave mean standard deviation of 2.72% (N = 11). The CV of the calibrations increases with calibration concentration. The sampling frequency can be adjusted to 0.1–10 s per sample, which is much more frequent than Ohira et al. [15]. The dynamic range is approximately from ~0 to 14 ppm, where saturation of the signal is observed, thus the difference in detection limit is influenced by the much broader dynamic range of the present system. It is expected that an LOD comparable to Ohira et al. [15] and a much better time resolution can be obtained by optimizing and adjusting UV led light, pumps flows, and reagent concentrations.

The linear relationship between the sensor output and the UV LED current makes it possible to adjust the dynamic range thereby changing the sensitivity. Having a flexible dynamic range could be an advantage under changing measurement conditions. For example, the UV LED current could be increased during measurement with a lower concentration compared to a pig house or a broader range could be used in order to determine the efficiency of an air cleaner by comparing inlet and outlet concentrations.

### 4.2. Pig House Measurements

The comparison between the fluorescence sensor and the CRDS analyzer shows high correlation between the two methods with a regression coefficient of 0.990. There is some variation in the CRDS data within a single 6-min cycle and the difference between the first and last concentration often exceed 100 ppb for the two highest concentrations, whereas the span of the low background concentration is within 10 ppb. The concentrations fluctuate over the day due to activities in the pig house and changing ventilation rate. The concentration obtained from CRDS only gives 3–4 data points each half hour compared to the continuous measurement with the fluorescence sensor, which makes it difficult to compare patterns on rapid time scale for the two sensors. However, it is very clear from Figure 6 that the fluorescence detector is capable of capturing relatively fast changes in concentration.

The CV of the fluorescence sensor and CRDS have similar magnitudes for measurements in the pig house see Table 1. Figure 5 shows good linearity with regression coefficients of 0.990 and the slope is close to 1.0 (slope = 0.97) and the intercept is 45, thus the biggest relative difference between the two methods are observed for concentrations close to zero as observed with the background measurement. The intercept is significantly different from zero (*p* < 0.05). The variation in concentration in the two room with pigs are caused by changes in ventilation rate, manure handling and other activities in the pig houses. Furthermore, the concentrations increase as the pigs grow.

The sensitivity in fluorescence intensity is affected by the pH concentration of the reagents, thus the reagents must be carefully prepared. From the flow rates, it is also clear that a considerable amount of reagent is needed for long-term measurements, and calibration should be conducted for each batch of reagents. Measurement of instrumental blank revealed no trends and only small variation between the different days. This suggests that there are no issues related to contamination of the reagents, thus the air intake through the charcoal filter removes NH_3_ efficiently. The design of the system is important to avoid contamination of the reagents over time. It must also be kept in mind that the OPA solution is light sensitive and should be kept dark to prolong the shelf life of the reagents.

The same pig house has shown significant emissions of organic acids [25], and some of these acids are known to be the cause of overestimation of NH_3_ concentration when measured by photoacoustic spectroscopy [19]. The CRDS is a suitable reference method as only negligible interference is observed when measuring in an agricultural environment [17], thus the results show no noticeable interference. This is also expected, as the fluorescence method is highly sensitive and selective for NH_3_ [11,15], and potential interference from organic amines are minimal as the concentration of these are low [26].

The performance of the fluorescence sensor shows a slightly higher LOD than CRDS (10 ppb vs. 0.3 ppb [17]), but it shows a comparable measurement range and response time. Inexpensive sensors such as electrochemical sensors have a 100–200 times higher LOD and a lower time resolution, thus these are more suitable for measurements above ~2 ppm [27]. Hale et al. [27] provide a summary of some low cost instruments with LODs typically in the low ppm range and accuracy from 3% to 25%. Electrochemical sensors use the electrolyte and require frequent calibration whereas instruments using optical absorption spectroscopy do not use any chemicals, e.g., CRDS. The use of chemicals is a disadvantage of the fluorescence sensor, especially because OPA is a toxic compound and waste must be handled according to the safety protocols. With the presented configuration of the instrument, approximately 0.3 L OPA solution (10 mM) is needed per day. The majority of the produced waste is ultrapure water containing absorbed NH_3_. This prototype can be optimized further regarding minimizing liquid flows to decrease the needed amount of chemicals.

The sensor has proven suitable for measurements of NH_3_ emissions from a livestock facility. However, from Figure 3 and the relatively low detection limits, it is anticipated that the sensor can also be used for sources characterized by lower concentrations. For example, the sensor may be applicable for micrometeorological mass balance measurements of emissions from open manure storage tanks [28]. Due to the relatively low cost and size, a number of sensors would potentially be able to measure at 4 heights and 2–3 locations with high time resolution compared to conventional off-line sampling systems with passive flux samplers that typically provide data on a daily to weekly basis [22,29].

## 5. Conclusions

The flow-based fluorescence sensor showed good agreement with standard gases during laboratory calibrations and direct comparison with a CRDS analyzer in a pig house. The regression coefficients were above 0.99 for both standard gas calibration and CRDS comparison. The results of the continuous measurements of NH_3_ gas with the fluorescence sensor in a pig house are promising for the use in an agricultural environment with many possible interfering compounds. Calibrations must be conducted with care on the reagents used in the actual measurements. The fluorescence sensor is made with simple and inexpensive components; thus, it is a proof of concept for an inexpensive sensor with great time resolution and linearity over a broad concentration range. Furthermore, the flexible setup makes it possible to change sensor settings as UV LED current and sampling time thereby providing flexible dynamic range and sensitivity.

## Figures and Tables

**Figure 1 sensors-21-01701-f001:**
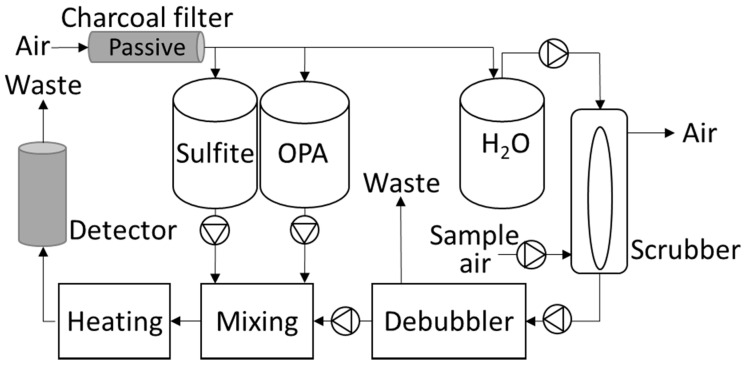
Schematics of the fluorescence sensor system with scrubber on the right and detector on the left. Six pumps handle the liquid and gas flow.

**Figure 2 sensors-21-01701-f002:**
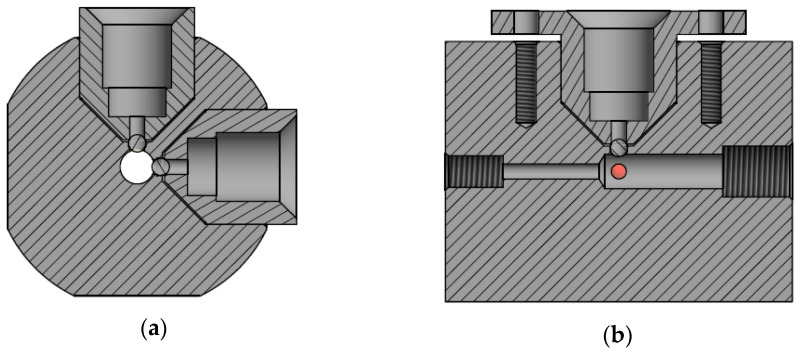
Polyether ether ketone (PEEK) flow cell with photo diode and LED installed in mounts. (**a**) shows an end view of a computer aided design (CAD) drawing of the flow cell. The two circles indicate the two ball lenses. (**b**) shows a side view of a CAD drawing of the flow cell. The two circles indicate the two ball lenses and water flow is from left to right.

**Figure 3 sensors-21-01701-f003:**
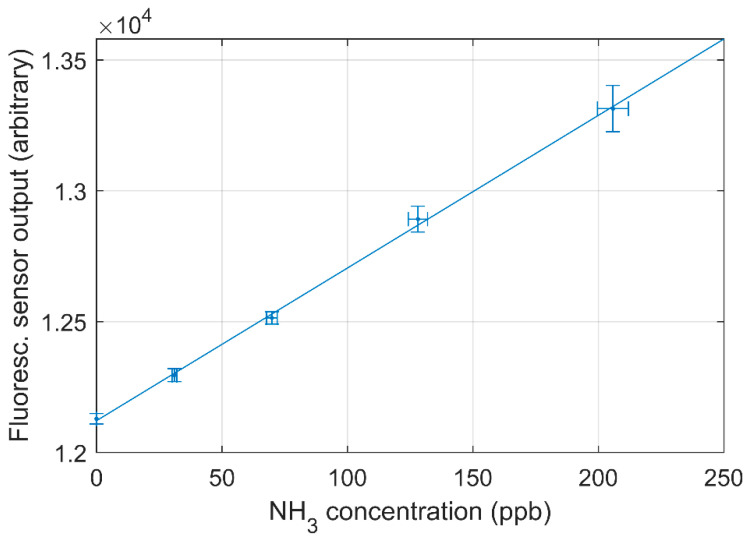
Laboratory calibration of the NH_3_ sensor with a dynamic dilution system of NH_3_ and N_2_ standard gasses. The blue line shows the linear regression line, y = 5.84x + 12121, R^2^ = 0.9990.

**Figure 4 sensors-21-01701-f004:**
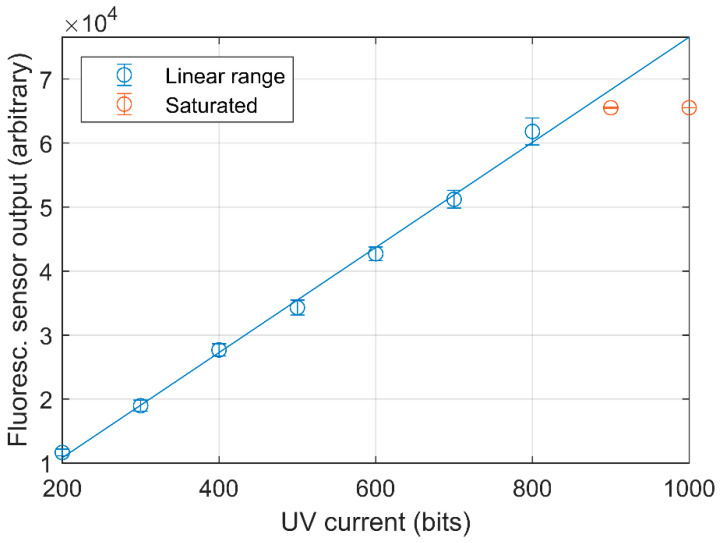
Fluorescence sensor output response to changes in UV LED current with a fixed NH_3_ gas concentration of 13.958 ppm. The blue line shows the linear regression line, y = 82.14x − 5591.1, R^2^ = 0.9965. The two red point show the saturated points, which are not included in the linear regression.

**Figure 5 sensors-21-01701-f005:**
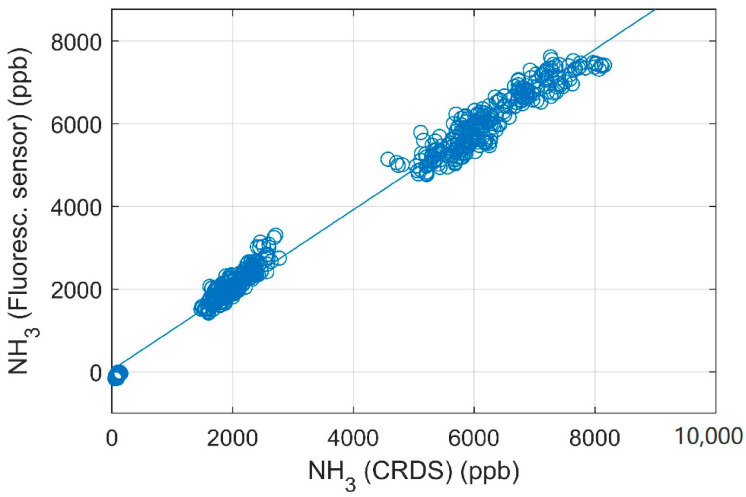
Linear regression between NH_3_ concentrations measured with a cavity ring-down spectroscopy (CRDS) analyzer and fluorescence detector in pig houses. The blue line shows the linear regression line, y = 0.97x + 45, R^2^ = 0.9903.

**Figure 6 sensors-21-01701-f006:**
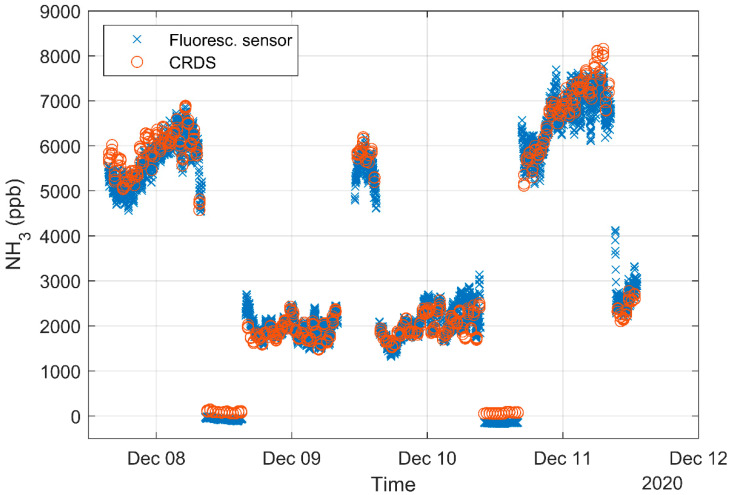
Minutely mean concentration for all fluorescence sensor data and only CRDS from the same position, i.e., two measurement cycles per hour for the CRDS.

**Table 1 sensors-21-01701-t001:** Mean, minimum, maximum, coefficient of variation (CV), and number of minutely mean measurements (N) for each sensor at each of the two pig houses when measuring at the same position.

Position	Sensor	N	Mean (ppb)	Min (ppb)	Max (ppb)	CV (%)
Room 1	CRDS	2176	6271	4572	8155	12.6
	Fluoresc. sensor	2176	6068	4539	7774	12.0
Room 2	CRDS	2288	1993	1477	2773	14.1
	Fluoresc. sensor	2288	2075	1318	4126	17.3

## Data Availability

The data in the study are available upon request to the corresponding author (jk@bce.au.dk).

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
