# Peer review of "Low-Cost Fluorescence Sensor for Ammonia Measurement in Livestock Houses"

_sensors, 2021, doi:10.3390/s21051701_

Round 1

Reviewer 1 Report

In this manuscript, A flow-based fluorescence sensor to measure ammonia gas was developed. I suggest for publication after minor revision.

1) The sensing mechanism (reaction) should be listed.

2)   How about the interference from other amines?

Reviewer 2 Report

The manuscript submitted by Kamp et al. presents low-cost fluorescence sensor for ammonia measurement in livestock houses. The work developed a low-cost sensor for continuous measurements of NH3 in the air of an agricultural environment with relatively high concentrations (> 10 ppm) based on absorption of NH3 in water and subsequent fluorescence detection. The fluorescence sensor is a proof-of-concept for a small, robust, flexible, and inexpensive sensor compared to other available measurement techniques. The sensor is subsequently tested for measuring NH3 in livestock houses. The paper present here is new and the work makes some sense, thus the manuscript is suggested for publication on Sensors after minor revision. Suggestions and questions are listed below:

  1. Please test the scrubber efficiency several times and list the experimental data in the article.
  2. In 2.5, how do you ensure that the ammonia concentration in the pig house does not change during the air extraction process?
  3. Can the fluorescent sensor continuously measure ammonia concentration in the pig house? It is recommended to add relevant experiments.

Reviewer 3 Report

The authors present an interesting work about a NH3 sensor based on fluorescence.

I will appreciate an answer of the authors to the following questions/suggestions:

  1. The authors should introduce a sentence to underline what kind of negative effects of ammonia on human health.
  2. In line 37 the authors say: ”Measurements of NH3 is of high importance to gain knowledge and ultimately to mitigate NH3 from different sources…”. I cannot see a clear connection between detection and mitigation.
  3. I think that the authors should show a picture of the real device.
  4. Did the authors test the LOD in laboratory?
  5. How did the authors avoid contaminations, for example by aminoacids, in the pig house?
  6. To improve the value of the work, the authors should compare the performances and the cost of the sensor with other devices.

Minor issues:

  • The first two paragraphs of the introduction should be rewritten to be more readable.
  • Line 48: what method?
  • In section 2.1 there is a change of the font.

Round 2

Reviewer 2 Report

The manuscript at the current version can be accpeted for publication.

Author Response

No comments in review report

Reviewer 3 Report

The authors clearly improved the manuscript and answered the questions. 

Author Response

No comments in review report